# Halotherapy—An Ancient Natural Ally in the Management of Asthma: A Comprehensive Review

**DOI:** 10.3390/healthcare9111604

**Published:** 2021-11-22

**Authors:** Radu Crisan-Dabija, Ioan Gabriel Sandu, Iolanda Valentina Popa, Dragos-Viorel Scripcariu, Adrian Covic, Alexandru Burlacu

**Affiliations:** 1Faculty of Medicine, ‘Grigore T. Popa’ University of Medicine and Pharmacy, 700115 Iasi, Romania; radu.dabija@umfiasi.ro (R.C.-D.); adrian.covic@umfiasi.ro (A.C.); alexandru.burlacu@umfiasi.ro (A.B.); 2Pulmonology Department, Clinic of Pulmonary Diseases, 700115 Iasi, Romania; 3Faculty of Material Science and Engineering, Tehnical University of Iasi Gheorghe Asachi, 700050 Iasi, Romania; ioan-gabriel.sandu@academic.tuiasi.ro; 41st Surgical Oncology Unit, Regional Institute of Oncology, 700115 Iasi, Romania; 5Nephrology Clinic, Dialysis and Renal Transplant Center, C.I. Parhon’ University Hospital, 700503 Iasi, Romania; 6Department of Interventional Cardiology, Cardiovascular Diseases Institute, 700503 Iasi, Romania; 7Romanian Academy of Medical Sciences, 030167 Bucharest, Romania

**Keywords:** halotherapy, hypertonic saline challenges, asthma, speleotherapy

## Abstract

The increasing production of modern medication emerges as a new source of environmental pollution. The scientific community is interested in developing alternative, ecological therapies in asthma. Halotherapy proved its benefits in asthma diagnosis, treatment, and prevention and may represent a reliable therapeutic addition to the allopathic treatment, due to its ecological and environment-friendly nature, in order to prevent or prolong the time to exacerbations in patients with asthma. We aimed to review up-to-date research regarding halotherapy benefits in asthma comprehensively. We searched the electronic databases of PubMed, MEDLINE, EMBASE for studies that evaluated the exposure of asthmatic patients to halotherapy. Eighteen original articles on asthma were included. Five studies in adults and five in children assessed the performance of hypertonic saline bronchial challenges to diagnose asthma or vocal cord dysfunction in asthmatic patients. Three papers evaluated the beneficial effects of halotherapy on mucociliary clearance in asthmatic adults. The therapeutic effect of halotherapy on acute or chronic asthma was appraised in three studies in adults and one in children. The preventive role was documented in one paper reporting the ability of halotherapy to hinder nocturnal asthma exacerbations. All studies seem to sustain the overall positive effects of halotherapy as adjuvant therapy on asthma patients with no reported adverse events. Halotherapy is a crucial natural ally in asthma, but further evidence-based studies on larger populations are needed.

## 1. Introduction

Current asthma guidelines provide clear protocols for escalating modern therapeutic agents (e.g., beta-agonists, anticholinergics, theophylline, or leukotriene modifiers). However, the increasing production and consumption of modern medication emerge as a new source of environmental pollution [1]. For instance, Theophylline was detected in ~1 μg/L concentration in United Kingdom river waters [2], and Salbutamol was detected in up to 5.4 ng/L in the drinking water in Serbia [3]. The European Medicines Agency acknowledged these threats in its last published guidelines on the environmental risk assessment of human medicines [4]. The scientific community became increasingly interested in developing or reviving alternative, more ecological therapies in asthma [5]. Moreover, the overprescribing of oral corticosteroids in children with asthma was demonstrated to be related to bone mineral density disorder and predisposition to fracture risk [6], as well as induced growth retardation [7]. Halotherapy (HT) may have a role as adjuvant therapy for the prevention of severe exacerbations, thus reducing the need for frequent pharmacological set-up interventions, especially in pediatric population.

Salt therapy in respiratory medicine started to be used in the 19th century due to its practical benefits on respiratory patients. This observation laid the foundation for speleotherapy (aerosols therapy provided in naturally occurring salt environments, such as salt-caves) [8,9] and HT (aerosols therapy in any salt-enriched environment, such as home devices or surfaces halo-chambers) [10].

HT has been associated with a plethora of benefits in asthma treatment and prevention [11]. Through an opposing mechanism, HT can also be used to diagnose asthma by bronchial challenge similar to physical exercise [12]. In addition, it has a bacteriostatic and positive effect on the local immune response. Despite its ecological, environment-friendly nature, and new and recent evidence, including on-going trials (NCT03556683), no scientific review covering the subject of HT benefits in asthma has been published in the last two decades [8]. Moreover, current asthma guidelines occasionally mention HT only as a bronchial challenge agent used to diagnose asthma [13,14,15].

Our aim is to (1) comprehensively review all studies evaluating the clinical utility and limitations of HT in asthma, (2) increase the awareness of the scientific community on the reliability of “old but efficient” natural and ecological therapies such as HT, targeting a more ecological medicine.

## 2. Materials and Methods

We searched the electronic databases of PubMed, MEDLINE, EMBASE for studies that evaluated the exposure of asthmatic patients to environmental therapies such as halotherapy or speleotherapy.

The terms used for searching were “asthma” plus each of the following: “halotherapy,” “halochamber,” “speleotherapy,” “salt mine,” “hypertonic saline aerosols.” The reference sections of relevant articles were also searched manually for additional publications. Observational studies, including prospective or retrospective cohort studies, RCTs, meta-analyses, guidelines, and case reports, were included if referring to this particular issue. Two independent reviewers selected studies by screening the title and abstract. Eighteen original articles on the uses of HT in asthma were included.

## 3. Diagnosis: Clinical Utility

Bronchial hyperresponsiveness (BH) in asthma is a condition in which the airway lumen narrows excessively in response to physical or chemical stimuli, such as HS (hypersaline) aerosols. In healthy patients, airways result in little or no caliber reduction when stimulated [16]. For this reason, HT has been used as a bronchial challenge agent for diagnosing asthma as an alternative to other airway stimulation agents such as direct mediators (histamine, methacholine) or physical exercise.

The mechanism by which saline aerosols generate bronchospasm is similar to asthma, as BH following osmotic stimulation indirectly reflects active airway inflammation. HS stimulation is a more specific approach for detecting asthma as it triggers initial inflammation that later leads to spasms of the bronchial smooth muscle cells [17,18,19]. In contrast, direct bronchial challenge agents such as histamine immediately bind to the receptors of the bronchial smooth muscle [20]. In addition to promoting the release of direct mediators such as histamine, leukotrienes, and prostaglandins [21,22], changes in the airway osmolarity caused by HS aerosols also lead to neural alteration, with an increase in the parasympathetic tone and release of neuropeptides [23].

Furthermore, HS is a more ecological, non-pharmacologic substance, rendering it preferable to pharmacologic agents such as histamine, methacholine, etc. This substance is low-cost, easy to prepare, and produced in a simple medical laboratory [24]. Compared to physical exercise, the challenge with HS requires less patient cooperation, cheaper equipment, and the airway response occurs gradually over time rather than abruptly as it does with cessation of exercise or hyperventilation [12].

### 3.1. HT—Bronchial Challenge for Asthma Diagnosis—Data from Past Literature

Several studies were conducted to compare the behavior and performance of different methods of BC. Such comparisons are of great interest to whether HS demonstrates to be similar in performance to the other BC techniques. It may prove to be a better alternative due to its advantages above. For instance, the bronchoconstrictor response to hypertonic aerosol and exercise was reported to be closely correlated [25,26].

In an adult population, 7.2% HS inhalations in controls did not show a significant fall of the forced expiratory volume in the first second (FEV_1_) compared to patients with bronchial asthma in which there was a highly significant reduction of 20% in FEV_1_ (chi-squared test: *X*^2^ = 75.42, *p* < 0.001), proving the ability of HS bronchoprovocation to diagnose asthma [27].

In children, a 4.5% HS challenge showed sensitivity and specificity similar to standardized exercise and pharmacologic challenges and higher sensitivity than cold air hyperventilation and distilled water to identify asthma (47% sensitivity and 92% specificity for the HS challenge and 46% sensitivity, and 88% specificity for the exercise test [28]). Challenge with 4.5% HS nebulized with the Timeter was found to have an accuracy of approximately 90% compared to 80% for hyperventilation and 90% for histamine in a group of children with mild asthma and control subjects [29]. Two non-comparative HS challenge studies found a sensitivity of 67% and a specificity of 90% [30] and a sensitivity of 61% and a specificity of 81%, respectively [31], for the diagnosis of asthma.

### 3.2. Newer Scientific Data

After a promising scientific perspective before 2000, there has been a gap in research studies. Interest in halotherapy research for asthma was overshadowed by physicians’ skepticism, lack of standardization, and insufficient research to back up the claim of safety and efficiency. However, paradoxically, confidence can only be built through more research.

Several papers emerged with more recent results and data. As such, 4.5% HS was compared to well-known osmotic agent mannitol as bronchial challenge agents combined with sputum induction. HS and mannitol challenges correlate with FEV_1_ maximal fall and the dose-response slope (DRS) curve [32]. However, in this study, the mannitol challenge had a higher sensitivity than the HS challenge to detect BH, contrasting with two other papers comparing bronchial challenge with mannitol versus HS [33,34]. Brannan et al. reported a 95.2% (91.1, 99.3) specificity and 65.1% (60.9, 69.3) sensitivity of response to HS compared to the clinical assessment of asthma and a 94.5% (89.9, 99.2) specificity and 59.8% (55.4, 64.2) sensitivity for mannitol compared to the clinical assessment of asthma.

Despite some differences, all three studies reported similar response rates and concluded that both HS and mannitol could be used to successfully induce sputum of good quality for analysis of inflammatory cells and inflammatory mediators for the assessment of asthma phenotypes (eosinophilic/non-eosinophilic asthma). This reveals another important use of bronchial provocation agents, as the eosinophilic asthma phenotype predicts an excellent response to corticosteroids [35] and associates to asthmatic exacerbations [36] but also to the new, more efficient treatments with monoclonal antibodies targeting the type 2 cytokines (IL-5, IL-4, IL-13, etc.) such as mepolizumab, reslizumab, or dupilumab [37]. Hence, identifying the asthma phenotype can positively impact therapeutic strategies.

In children, 4.5% HS was used as a bronchial provocation test in diagnosing asthma in children with chronic recurrent cough (CRC), achieving a sensitivity and specificity of 84.1% and 100.0%, respectively [20]. Most of the patients with CRC in this study (98%) were diagnosed with asthma compared to the studies of Gunadi [38] and Rahajoe et al. [39], in which the diagnosis of asthma was based on clinical examination only and found the asthma prevalence among CRC patients ranged between 62.5% and 64.5%. This difference may account for another significant benefit of the HS challenge test.

Even though few studies were available for the usefulness of HS as a bronchial challenge agent, HS was referred to in three important last-hour consensus guidelines. Among the asthma guidelines with updated versions available, the European Respiratory Society clinical practice guidelines for the diagnosis of asthma in children aged 5–16 years [15], the strategy for asthma management and prevention of the Global Initiative for Asthma (GINA) network [14], and the British guidelines on the management of asthma (BG) [13] mentioned HS as an alternative to bronchial provocation. It is worth highlighting that the BG achieved the highest score with the Appraisal of Guidelines for Research and Evaluation II (AGREE II) Instrument [40]. Considering the present-day attention given to HS by highly rated consensus reports, there must be less room for skepticism and a greater openness to intensifying research and standardization.

### 3.3. Hypertonic Saline Challenge in the Diagnosis of Vocal Cord Dysfunction in Patients with Asthma

Airway challenge with HS is used worldwide as a gold standard in diagnosing asthma as there is evidence that links airway hyperresponsiveness to HS with underlying airway inflammation in children with asthma rather than histamine [41] or other stimuli [42]. Yet, another surprising benefit of bronchial challenge with HS was discovered by Reay et al. [43]. Changes in forced inspiratory flow and visual analog scores (VAS) after HS challenge in 27 asthmatic patients discriminated between patients with vocal cord dysfunction (VCD) and those with asthma alone and healthy controls. After further testing and validation, this instrument may prove helpful in the diagnosis and treatment of VCD.

## 4. Protective Effects in Asthma

In addition to the bronchostimulation effects used in the diagnosis of asthma, in a seemingly contradictory way, alternative molecular pathways triggered by HT may also have protective and preventive repercussions in asthma, as reported [44].

### 4.1. Improving Mucociliary Clearance

Mucociliary clearance (MCC) is a vigorous innate defense to remove mucus and related toxins, germs, viruses, and inflammatory cells from the respiratory airways. As airway epithelial cultures become increasingly dehydrated, they lose their ability to transport mucus, establishing a critical link between mucus dehydration and deficient MCC [45].

There is evidence that the mucus obtained from asthmatic patients is dehydrated more than expected [46], and defective MCC is present in moderate-severe asthma, especially during acute exacerbations [47]. Studies have shown that HS improves mucus hydration and rheology [11,48] by disrupting mucus ionic bonds (which leads to the lowering of the mucus viscosity and elasticity) [49]. In addition, HS improves MCC by reducing the thread-forming ability of sputum (as shown in cystic fibrosis) [50] and by dissociating deoxyribonucleic acid from the mucoprotein (which results in the mucoprotein being digested by the natural proteolytic enzymes) [51].

A small group of well-controlled, moderate-severe female asthmatic patients was treated with a single test dose of albuterol (four puffs by metered-dose inhaler) followed by HS (7% sodium chloride, 4 mL) [11]. Mean ± standard deviation clearance over 60 min of dynamic imaging was 8.9 ± 7.9% (baseline) versus 23.4 ± 7.6% (acute HS) (*p* < 0.005). However, this enhancement was not maintained over a 4-h period where post-HS treatment clearance was 9.3 ± 8.2%. Although these results do not support the use of aerosolized HS for maintenance therapy, as the long-term effects are considered uncertain by the authors, these findings suggest therapeutic benefits during acute exacerbations for patients with moderate to severe asthma. No decline in lung function was seen up to 30 min after therapy.

Albuterol treatment was included before HS treatment in the reported study to reduce the risk of airway hyperreactivity. Therefore, the treatment effects on MCC were reported for the combination of albuterol and nebulized HS. β-agonists have been shown to acutely improve MCC in asthma [52], albeit the relative impact of HS versus albuterol could not be determined. However, β-agonists enhance MCC to a lesser extent than observed in the reported study [11].

The rest of the effect must have been attributable to HS. In support of this statement, the effect on MCC administering ultrasonically nebulized 14.4% saline in asthmatic patients without prior delivery of β-agonists was measured [43]. MCC of the whole right lung in 1 h was 68 ± 10% with 14.4% saline vs. 44 ± 14% with 0.9% saline. Although the hypertonic aerosol significantly increased MCC, all asthmatic patients presented a fall of FEV_1_ higher than 15%.

Moreover, the safety of induced sputum with HS inhalations without prior β-agonists was assessed in asthmatics within 24 h of two commonly used airway challenges: endotoxin and dust mite allergen. HS inhalations have proven safe with no significant lung function decline before or after inhaled challenges with endotoxin or dust mite allergen [53]. On top of that, the induced sputum significantly enhanced MCC rates before and after inhaled endotoxin challenge [53].

### 4.2. Therapeutic Effect of Halotherapy on Acute and Chronic Asthma

In addition to mucus rehydration and clearance, additional postulated mechanisms may contribute to the therapeutic effect in asthmatic patients. HT can break the ionic bonds within the mucus gel, thereby reducing cross-linking and entanglements [53], stimulate cilial beat [54], reduce the airway wall edema by absorbing water from the surrounding tissues [32], and trigger sputum induction and cough to improve airway obstruction [55].

Two studies assessed the veracity of the therapeutic benefits of HT in acute asthma. In a double-blind, randomized clinical trial of 340 adult patients with acute asthma attacks, the treatment with 3% HS plus salbutamol resulted in a significant increase compared with solely salbutamol in both peak expiratory flow rate (PEFR) and FEV_1_ at 40 min (0.11 ± 1.36; *p* = 0.036 and 0.05 ± 1.16; *p* = 0.033, respectively) and 60 min (0.15 ± 1.12; *p* < 0.001 and 0.11 ± 1.22; *p* = 0.011, respectively) after treatment [56]. No severe adverse events were recorded. The study showed the short-term efficacy of 3% HS in acute asthma attacks. Conversely, the long-term efficacy was demonstrated in another double-blind, randomized trial, where the use of a dry-salt inhaler 20 min/day versus placebo in stable asthma showed an overall improvement of the FEV_1_, forced vital capacity (FVC) by 4%, and PEFR by 25% after four months of treatment while using HT as supplemental, adjuvant therapy along with the chronic asthma treatment [57].

Another combination therapy was proposed, including recreational winter exercise and speleotherapy in highly functioning asthma patients with reasonable disease control [58]. No significant effects were found for any spirometry parameter. However, for the speleotherapy combined with the exercise group (compared to the exercise alone group), relative treatment effects indicated a more substantial decrease of white blood cell and nasal eosinophilic cell count, suggesting a sustainable improvement effect of the mucociliary clearance time and post hoc test showed a trend in the interaction effect at day 60 for allergic symptoms, revealing a faster return to baseline within the speleotherapy group. However, according to the relative treatment effects, the decrease of fractional exhaled nitric oxide (a surrogate parameter to assess allergic airway inflammation) [59] was reported in the exercise group (Ex. × T0 = 0.57, Ex. × T1 = 0.34) compared to the speleotherapy combined with exercise group (Sp. × T0 = 0.50, Sp. × T1 = 0.47). No participant experienced adverse effects during the winter sports and speleotherapy treatment. However, the study’s impact was limited by the small sample size and the fact that the included population presented good baseline spirometry.

In children, administering nebulized albuterol containing HS 5% to wheezing preschool children presenting to the emergency department significantly shortened the length of stay (a median of 2 days compared with the typical saline group median of 3 days, *p* = 0.027) and hospital admission rate (62.2% in the hypertonic saline group versus 92.0% in the standard saline group, *p* ≤ 0.05) with no documented side effects [60]. An important finding to consider is that 83% of the sputum samples were positive for one or more respiratory viruses, with the most common being rhinovirus. Rhinovirus is the most common cause of acute wheezing episodes in preschool children [61], leading to decreased extracellular adenosine triphosphate levels, which in turn cause airway surface liquid dehydration [62]. Additionally, the submucosal edema and inflammation caused by the infection hamper mucus clearance. In this context, HS aerosols are indeed one of the most suitable therapeutic choices for their ability to improve mucus hydration and clearance.

### 4.3. Impact on Bacterial Infections in Asthma

Atypical bacterial play a potential role in inducing and exacerbating asthma [63]. Early studies involving *Chlamydia pneumoniae* suggest a link to infection and the onset of asthma [64]. Therefore, if a bacterial infection is suspected, therapy should be directed toward the identified microorganisms.

Saline aerosols appear to have potent antimicrobial effects. Studying the phagosomal acidification during leukocytes bacterial internalization, Hackman et al. in 1997 [65] and then later Westman and Grinstein [66] revealed that an essential role of microbial growth inhibition was played by the ability of the cell to lower the pH by activating the Na^+^/H^+^ antiporter for which, the increased inward Na^+^ gradient drives out, through the ATPase, the H^+^.

Activated neutrophils experience a burst of oxygen consumption, consequent to the behavior of their NADPH-oxidase and generate oxidants in their phagosomes or the extracellular space, acting as an electron transferase and thus generating a superoxide anion [67]. This superoxide anion exerts minimal antimicrobial effect, but it is rapidly converted to hydrogen peroxide (H_2_O_2_), which, under the catalyzation of myeloperoxidase (MPO), is then converted to a very potent antimicrobial agent, the hypochlorous acid [68].

These data suggest that, firstly, the presence of Na^+^ in the extracellular space activates the Na^+^/H^+^ ATPase that drives out the H^+^ and thus lowers the pH, a process called phagosomal acidification, creating an inhibitory medium for bacterial growth. In addition, that the abundant presence of chloride ions precipitates the formation of hypochlorous acid, a significantly more powerful antimicrobial agent than hydrogen peroxide, especially against *Escherichia coli* [69] or *Klebsiella pneumoniae* [70].

## 5. Prevention of Asthma Crisis

Although the inhalation of HS causes bronchoconstriction in asthmatic subjects, the repeated challenge of the airways with HS results in loss of airway responsiveness, known as the refractory period [71]. The existence of the refractory period paves the way for the use of HT in the prevention of asthma exacerbations.

The impact of refractoriness was evaluated on patients with nocturnal asthma (defined by an increase in symptoms, need for medication, airway responsiveness, and/or worsening of lung function, usually occurring from 4 to 6 a.m.) [71]. The mean overnight reduction of FEV_1_ was significantly higher (523 ± 308 mL or 22.75 ± 15.40%) in the control day (with no challenge in the afternoon) compared to 206 ± 414 mL or 9.81 ± 17.42% on the study day (with two HS challenges in the afternoon), during the refractory period (*p* = 0.021). Refractoriness following HS challenges efficiently prevents nocturnal worsening of asthma. The authors speculate that the duration of the refractory period following HS challenges is at least 10 h [71]. However, no scientific evidence to date demonstrated its duration.

## 6. Limitations and Adverse Events of Halotherapy

A few uncertainties and unresolved issues overshadow the promising prospects.

Firstly, the small number of participants in the studies limits the generalizability of conclusions. Except for the study of Brannan et al. [34] that comprised 592 subjects, including both children and adults, the sample sizes of the other 19 included studies ranged from 8 to 340 in adults or 19 to 393 in children. Although the included studies reported rapid, spontaneous, and well-tolerated recovery with no need for a bronchodilator in asthmatic patients treated with HT, the total small number of the patients included in the studies cannot convincingly dispute the issues related to the safety of employing challenge tests such as the risk of inopportune bronchospasm with possible perilous consequences and the risk of late asthmatic reactions.

Secondly, with very few exceptions, most studies only evaluated short-term efficacies of HT in asthma patients. A recent study on the effects of HS treatment in chronic bronchitis showed slowing of MCC relative to baseline after multiple treatments over two weeks [72]. It is compulsory to reveal whether a longer-term effect may also occur in asthma.

Consequently, the current guidelines only incorporate references to HT as alternatives to bronchial challenge for diagnosing asthma, yet do not include any recommendations regarding the therapeutic or preventive effects of HT in asthmatic patients.

## 7. Conclusions

Modern respiratory medicine is fully aware of the limitations of salt aerosols in treating pulmonary disorders, but supplementary to any inhalation-delivered medicine, HT has been proven an important natural ally. Despite the skepticism of some, HT demonstrated to be a complex, reliable, cost-effective solution in the diagnosis, treatment, and prevention of asthma and a much-needed ecological alternative to present-day medicines prone to environmental risk. To fully tackle current skepticism and pave the way to systematic inclusion in the guidelines, more evidence-based studies on larger populations are imperatively needed.

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
