# Peer review of "Halotherapy—An Ancient Natural Ally in the Management of Asthma: A Comprehensive Review"

_healthcare, 2021, doi:10.3390/healthcare9111604_

Round 1

Reviewer 1 Report

Aim of the review was up-to-date research regarding halotherapy benefits in asthma . Actually 10 out 20 included studies which assessed the diagnostic value of hypertonic saline  bronchial challenges and not the efficacy of halotherapy ! Authors should not consider these studies or they should separate the two aspects of saline inhalation, i.e.  the diagnostic value of hypertonic saline bronchial challenge and halotherapy in asthma.

Abstract . The first statement “The increasing production of modern medication emerges as a new source of environmental pollution” is not supported by the evidence and it seems an inappropriate sentence to introduce the topic of the review. In the last years the biological therapy for T2 asthma has changed the therapeutic approach to asthma therapy, with important steroid sparing effect.

The second statement “The scientific community is interested in developing alternative, ecological therapies in asthma” is arbitrary. Which scientific community are the Authors referring to? The salbutamol concentration found in the water is ridiculous and it could not be changed simply by combining salbutamol with hypertonic saline !

Pag 3 lines 135-6 Authors should add that the eosinophilic asthma phenotype predicts not only an excellent response to corticosteroids, but, if asthma is severe, it may be treated with biological therapy, such as anti IL-5 monoclonal antibodies or dupilumab.

Pag.4 lines 182-4 It is not supported by the results of the reported study (ref 7) what Authors suggest, that is therapeutic benefit in moderate to severe asthma patients with acute exacerbations !

Pag 5 lines 211-21 Authors report one study (ref 49) aimed to compare the efficacy of 3% hypertonic saline (HS) plus salbutamol with solely salbutamol on management of acute adults' asthma based on peak flow meter findings. The increase in PEFR and FEV-1 obtained by the combination of salbutamol and HS is trivial, although statistically significant. In acute asthma the important outcome are the time spent in emergency room, the need to admit patients in Hospital, as well the need to intubate the patients. This study did not show any short-term efficacy of 3% HS in acute asthma attacks, but simply a trivial increase in PEFR !

I was not able to evaluate the other study (ref 50) aimed to evaluate the long-term efficacy of HS, as the reference is incomplete.

Conclusions. Authors correctly conclude at the end of the manuscript that more evidence-based studies on larger populations are imperatively needed. On the other hand, in the Abstract they conclude that all studies reveal apparent (what does it means? it is not scientifically sound!) positive effects on asthma patients  with no reported adverse events.

Reviewer 2 Report

To the Authors

Summary

Indeed, the high prevalence of asthma in adults and children is a growing concern worldwide. This condition is associated with increased morbidity, in some cases mortality, poor quality of life for the patient and family along with increased economic costs for the national healthcare system. To date, there is no cure for asthma. Symptoms can only be controlled by medication which on a long-term basis may cause side effects. Furthermore, adherence to pharmacological therapy is sub-optimal, especially in children. Therefore, it is worth exploring non-pharmacological and ecological-friendly approaches.

This study was a systematic appraisal of the literature evaluating the clinical utility as well as the limitation of Salt Therapy in asthma. Based on the evidence outlined, it appears that salt therapy may be a cost-effective, non-pharmacological alternative or at least adjunct to conventional medication therapy. The authors conclude the need for more well-designed large studies in order to elucidate the therapeutic benefits of salt therapy in the management of asthma as well as in the prophylaxis of future exacerbations.

Given the burden of asthma and the issue of poor patient adherence to pharmacological therapy, there is a need for more research on alternative means in controlling and preventing future asthma exacerbations.

Reviewer 1

Comments to the Authors

Overall, this review was well-written, concise, and comprehensible.

However, there are minor typographical errors, and according to the author guidelines, the citing of journal titles needs to be amended.

Please refer to my concerns below.

-Line 44 ‘The scientific community became increasingly interested in developing or reviving alternative, more ecological therapies in asthma’

I think it is worth mentioning that apart from the environmental pollution, most importantly, the detrimental effects of long-term use of medication in children. For example oral cortisone and stunted growth in children etc

Thus, emphasizing the importance of non-pharmacological approaches in the management of asthma symptomology, especially in the pediatric population.

-Line 124. Throughout the manuscript, FEV1 has been incorrectly written.

Please correct  FEV1 to FEV1.

-Line 139, ‘In children, 4.5% HS was used as a bronchial provocation test in diagnosing asthma’.

Why is “In’ in bold text?

-Line 141 ‘98% of patients with CRC in this study were diagnosed with asthma compared to the studies of Gunadi et al. [34] and Rahajoe et al.’

Grammatically, numbers (98%) should not start a sentence.

Perhaps, ‘Approximately 98% of patients with CRC………

-Line 215. 0.11±1.36; p=0.036,

 In scientific writing statistical data should be presented in APA style as follows.

Not p<0.001, 0.11±1.36; p=0.036, but written as 0.11 ± 1.36; p = 0.036, p < 0.001

That is, including a space before and after  ± and = , as well as a space after the p-value and before <.

-Line 266 Not H202 but H202 with subscript 2

-In the reference section. As set out in the author guidelines, journal titles should be abbreviated and having the first letter of titles capitalized. When uncertain refer to PUBMED.

For example :

Not Journal allergy and clinical immunology but J Allergy Clin Immunol.

Not American journal of respiratory and critical care medicine but Am J Respir Crit Care Med

Not The European Respiratory Journal but Eur Respir J etc

Reviewer 3 Report

General comments

This review comprehensively describes halotherapy and speleotherapy as alternatives to existing asthma diagnosis and treatment. This is a valuable review summarizing several reports on halotherapy and speleotherapy. However, some sections require modification.

Major comments

  1. The problem of halotherapy and speleotherapy is lack of scientific evidence in the regulation of airway inflammation and bronchoconstriction. In section 4-3, the authors concisely refer to the mechanism of beneficial effects of saline inhalation. However, in the rest of parts, only the outcomes of halotherapy in clinical studies are described. The authors should explain about the molecular mechanism of halotherapy in each section or make a new section for this point.
  2. In line 40-41: The authors describe the environmental pollution of theophylline and salbutamol. Is the concentration (theophylline:~1 μg / L, salbutamol: 5.4 ng / L) actually harmful to the body?
  3. In line 157: The authors describe clinical utility of hypertonic saline in differential diagnosis between vocal cord dysfunction and bronchial asthma. It should be described more scientifically why it was possible to distinguish two diseases by hypertonic saline.
  4. In line 168: The title of section 4 is “the protective effect in acute asthma”. Despite this title, the section 4-2 involves the therapeutic effect of halotherapy on chronic asthma. Thus, the title of section 4 should be reconsidered.
  5. Section 4-3: The section 4-3 explains about anti-microbial effects of halotherapy in chronic airway inflammation. However, the title name is about diagnosis of asthma and vocal cord dysfunction. The authors should correct this point by renaming the title.

Minor comments

  1. Line 252: Please change the word “the diseases” to “asthma”.
  2. Line 287: Please change the word “ten” to “10”.
  3. Line 287: Please change the word “h” to “hours”.

Reviewer 4 Report

Objective:

In the current review, the authors focus on the effectiveness of hypertonic saline as an ecological therapy in asthmatic patients, i.e., in asthma diagnosis, prevention and treatment. After an extensive literature search in the most appropriate electronic databases, the authors selected 20 relevant papers to summarize the most important results.

As a consequence, hypertonic saline and halotherapy were reported to be valuable tools in the diagnosis of asthmatic patients, protective, e.g., by the enhancement of the mucociliary clearance, as well as being a preventive measure of asthma crisis.

The authors point to the current guidelines that solely incorporate references to hypertonic saline as alternatives to bronchial challenge for the diagnosis of asthma but do not include any recommendations regarding the therapeutic or preventive effects in asthmatic patients. Therefore, they invoke clinical studies to get additional data for the evaluation and application of this cheap and effective treatment.

Points of criticism:

  1. Abbreviations that need to be explained at their first mention, e.g., “HT” or “HS”.

  1. In the Abstract the authors refer to 20 original articles on asthma and indicate 5 studies on adults, 5studies on children for the diagnosis of asthma or vocal cord dysfunction. Further, 3 papers evaluated the beneficial effects of HT and mucociliary clearance. Another 3 papers for the therapeutic effect of HT as well as 1 publication in children. 1 paper reported the preventive role of HT to hinder nocturnal asthma exacerbations. In sum these are 18 publications – what about the remaining 2 publications?

  1. Several typographical errors need to be improved – predominantly “space” and case sensitivity.

  1. On page 5 – line 223 the authors devote the term “speleotherapy”. Presumably, they mean a spa therapy in a salt mine – this should be drafted more precisely.

  1. References need to be improved distinctly, i.e., wrong positioning or even missing the name of the Journal, e.g., Ref. 53 and 50. A severe mistake occurred in Ref 44 and 45, i.e., Ref 45 is a repetition of Ref 44. That needs to be revised also in the manuscript.

  1. In times of COVID-19, several papers were published dealing with the protective effect of hypertonic saline as a prevention measure in combination with filtering facepiece. It would be worthwhile to create a subchapter to focus on the anti-microbial effects concerning yeast (Tatzber F. et al. Prev Med Rep 20 (2020) 101270 https://doi.org/10.1016/j.pmedr.2020.101270), bacteria (Rubino et al. Sci Rep 10 (2020) 13875 https://doi.org/10.1038/s41598-020-70623-9) and viruses (Tatzber et al. Int J Environ Res Public Health 2021, 18, 7406; https://doi.org/10.3390/ijerph18147406). This would increase the significance of the current manuscript.

Round 2

Reviewer 3 Report

The authors have addressed all my comments and I have no further critiques. 

Reviewer 4 Report

The proposals for improvement have largely been implemented. Unfortunately, the authors did not take the last point into account, i.e., the antimicrobial effect of hypertonic salt solutions on filtering face-piece. I regret this procedure, but I understand their hesitant approach, although the protective effect in connection with FFP´s has been proven and would have increased the significance of the work even more.

Author Response

We are grateful for all the useful remarks and we thank the Reviewer for the effort and time spent to make a qualitative review of our paper.

Regarding the last point from the Reviewer’s observation concerning the inclusion of a sub-chapter about antiviral effects of halotherapy in COVID-19, the current level of evidence is very weak and we firmly believe that it is wiser to not yet speculate about such a sensitive topic. Given the tendency of some members of the community towards exclusive alternatives to vaccination and prevention, speculating about a positive effect of alternative therapies such as hypertonic saline could do more harm than good (in this particular context).

However, with the accumulation of new and stronger evidence, we assure the Reviewer that a future review will be taken into account by this team. We believe that the idea of assessing whether halotherapy might have a role in managing or preventing COVID-19 is definitely worth of consideration.